# Baseline and acquired resistance to bedaquiline, linezolid and pretomanid, and impact on treatment outcomes in four tuberculosis clinical trials containing pretomanid

Juliano Timm[1]*, Anna Bateson[2], Priya Solanki[2], Ana Paleckyte[2], Adam A. Witney[3], Sylvia A. D. Rofael[2,4], Stella Fabiane[5], Morounfolu Olugbosi[6], Timothy D. McHugh[2], Eugene Sun[1]

1 TB Alliance, New York City, New York, United States of America, 2 Centre for Clinical Microbiology, University College London, Royal Free Campus, London, United Kingdom, 3 Institute of Infection and Immunity, St George's, University of London, London, United Kingdom, 4 Faculty of Pharmacy, University of Alexandria, Alexandria, Egypt, 5 MRC Clinical Trials Unit at University College London, London, United Kingdom, 6 TB Alliance, Pretoria, South Africa

* juliano.timm@tballiance.org

**Data Availability Statement:** All relevant patient clinical data is provided in the manuscript's main

## Abstract

Bedaquiline (B), pretomanid (Pa) and linezolid (L) are key components of new regimens for treating rifampicin-resistant tuberculosis (TB). However, there is limited information on the global prevalence of resistance to these drugs and the impact of resistance on treatment outcomes. *Mycobacterium tuberculosis* (MTB) phenotypic drug susceptibility and whole-genome sequence (WGS) data, as well as patient profiles from 4 pretomanid-containing trials–STAND, Nix-TB, ZeNix and SimpliciTB–were used to investigate the rates of baseline resistance (BR) and acquired resistance (AR) to BPaL drugs, as well as their genetic basis, risk factors and impact on treatment outcomes. Data from >1,000 TB patients enrolled from 2015 to 2020 in 12 countries was assessed. We identified 2 (0.3%) participants with linezolid BR. Pretomanid BR was also rare, with similar rates across TB drug resistance types (0–2.1%). In contrast, bedaquiline BR was more prevalent among participants with highly resistant TB or longer prior treatment histories than those with newly diagnosed disease (5.2–6.3% vs. 0–0.3%). Bedaquiline BR was a risk factor for bacteriological failure or relapse in Nix-TB/ZeNix; 3/12 (25%, 95% CI 5–57%) participants with vs. 6/185 (3.2%, 1.2–6.9%) without bedaquiline BR. Across trials, we observed no linezolid AR, and only 3 cases of bedaquiline AR, including 2 participants with poor adherence. Overall, pretomanid AR was also rare, except in ZeNix patients with bedaquiline BR. WGS analyses revealed novel mutations in canonical resistant genes and, in 7 MTB isolates, the genetic determinants could not be identified. The overall low rates of BR to linezolid and pretomanid, and to a lesser extent to bedaquiline, observed in the pretomanid trials are in support of the worldwide implementation of BPaL-based regimens. Similarly, the overall low AR rates observed

body and S1 Appendix. Notably, the whole-genome sequence data have been deposited in the European Nucleotide Archive (https://www.ebi.ac. uk/ena/browser/home), and all ENA accession numbers are listed in S1 Appendix.

**Funding:** This work was supported by TB Alliance with funding from Australia's Department of Foreign Affairs and Trade, the Bill & Melinda Gates Foundation, Germany's Federal Ministry of Education and Research through KfW, Irish Aid, Netherlands Ministry of Foreign Affairs, United Kingdom Department of Health, United Kingdom Foreign, Commonwealth and Development Office, and the United States Agency for International Development. The funders had no role in study design, data collection and analysis, decision to publish, or preparation of the manuscript.

**Competing interests:** The authors have declared that no competing interests exist.

suggest BPaL drugs are better protected in the regimens trialed here than in other regimens combining bedaquiline with more, but less effective drugs.

## Introduction

Tuberculosis (TB) remains a global health crisis, responsible for 1.6 million deaths and 10.6 million new cases globally in 2021, of which half a million corresponded to rifampicin-resistant (RR) or more resistant forms of the disease [1]. The established treatment regimens are clearly inadequate to control the pandemic. Fortunately, the 21st Century saw the development of novel drugs: the diarylquinoline bedaquiline (B) and the nitroimidazoles pretomanid (Pa) and delamanid. Particularly bedaquiline and pretomanid, along with the repurposed oxazolidinone linezolid (L), appear as key components of new therapies for TB. Regimens such as BPaL and BPaLM (BPaL plus moxifloxacin) have been recently endorsed by the World Health Organization (WHO) for the treatment of multidrug resistant TB (MDR-TB; defined as combined resistance to rifampicin and isoniazid) and RR-TB [2], while others are currently under investigation in clinical trials involving both drug-sensitive (DS) and drug-resistant (DR) TB participants, as summarized in the New TB Drugs Clinical Pipeline [3].

However, the promise of more efficacious, shorter treatments based on the BPaL backbone (or with next-generation diarylquinolines and oxazolidinones replacing bedaquiline or linezolid, respectively) [4], is challenged by our limited knowledge of the prevalence of resistance to these classes of drugs and the impact of acquired resistance (AR) on the efficacy of the different regimens. This is mainly due to the limited capacity for phenotypic drug susceptibility testing (DST) for these drugs, even in high-income countries [5], as well as the complex genetics of resistance to bedaquiline and pretomanid/delamanid, which has hindered the development of rapid molecular tests. In consequence, much of the published data is limited to bedaquiline and in a small number of countries. Most notably, a 5-year (2015–2019) bedaquiline resistance surveillance study involving 5036 MDR-TB isolates from bedaquiline-treatment-naïve patients across 11 countries revealed significant differences in bedaquiline-resistance rates between countries–e.g., 3.4% in South Africa vs. 0.2% in India–and between TB resistance types–e.g., 2.4% in XDR-TB versus 1% in MDR-TB–which remain unexplained [6]. This study also examined linezolid resistance and found an overall rate of 1.5%. In another study focusing exclusively on South Africa (2015–2019) [7], a cross-sectional surveillance analysis of 2023 DR-TB patients pointed to high bedaquiline baseline resistance (BR) rates: 3.6% among those with no previous exposure or unknown previous exposure to bedaquiline or clofazimine, and 21.1% among those with previous exposure. The longitudinal arm of the same study showed that 16/695 (2.3%) patients analyzed acquired bedaquiline resistance during treatment. Other, smaller patient cohorts treated with individualized bedaquiline-based regimens exhibited similar or higher rates of bedaquiline AR [8, 9].

Bedaquiline was approved as part of combination therapy for MDR-TB by the U.S. and European regulators in 2012–2014 [10]. As a result of WHO endorsement, bedaquiline has now been used in >100 countries and constitutes the basis of all-oral MDR/RR-TB regimens endorsed by WHO [2]. It binds to subunit C of the mycobacterial ATP synthase, blocking energy production [11]. At least 6 *Mycobacterium tuberculosis* (MTB) genes (*atpE*, *mmpR5* [*Rv0678*], *mmpL5*, *mmpS5*, *pepQ* and *Rv1979c*) are implicated in resistance to bedaquiline *in vitro* or in mice [12]. In clinical settings, however, most phenotypically confirmed bedaquiline-resistant cases have been linked to mutations in *mmpR5*, the transcriptional regulator of

the *mmpL5/mmpS5*-encoded transporter which can pump bedaquiline out of the tubercule bacilli [13]. MmpL5/MmpS5 is also active on another anti-TB drug, clofazimine, hence *mmpR5* mutations often lead to bedaquiline/clofazimine cross-resistance [13]. To date, few clinical bedaquiline-resistant cases have been attributed to mutations in the ATP synthase gene, *atpE*, which result in higher-level resistance [6–9, 14].

Linezolid, originally licensed for treatment of certain infections caused by Gram-positive bacteria, has been increasingly incorporated in regimens for MDR- and extensively drug resistant TB (XDR-TB; defined, prior to 2021, as MDR-TB with additional resistance to a fluoroquinolone and a second-line injectable) [15]. It binds to the 50S ribosomal subunit and blocks translation [16]. Mutations in the *rplC* and *rrl* genes, which encode the ribosomal protein L3 and 23S rRNA, respectively, are known to result in linezolid resistance [17, 18]. These mutations are very rare in clinical settings, probably due to their impact on mycobacterial fitness [18].

Pretomanid was approved by the U.S. and European regulators in 2019–2020 for use, as part of the BPaL regimen, to treat XDR-TB and treatment intolerant or non-responsive MDR-TB. Pretomanid kills actively replicating MTB under aerobic conditions by inhibiting mycolic acid biosynthesis and blocking cell wall synthesis, and non-replicating MTB under anaerobic conditions by respiratory poisoning via nitric oxide release [19, 20]. These activities require nitro-reduction of the drug within the mycobacterial cell, which is mediated by the products of the *ddn*, *fbiA-D*, and *fgd1* genes. Mutations in any of these 6 genes have been associated with pretomanid resistance, at least *in vitro* or in mice [17, 21]. In clinical settings, however, there is still little information on the prevalence and impact of the many genetic variants identified. This is also true for delamanid, which has been in use for the treatment of MDR-TB since 2014 [22], and shares the same activation pathway and hence exhibits partial cross-resistance with pretomanid [17].

Here we analyzed phenotypic DST, whole-genome sequence (WGS), and patient clinical data from 4 TB Alliance pretomanid-containing trials (STAND [23], Nix-TB [24], ZeNix [25] and SimpliciTB [26]), representing a combined dataset of >1,000 TB patients enrolled between February 2015 and March 2020 in 12 countries (Table 1), to determine the BR and AR rates to bedaquiline, linezolid and pretomanid and their impact on treatment outcomes; as well as to study the genetic determinants and risk factors underlying these resistances. Importantly, our study describes the largest cohort of patients with pretomanid susceptibility data reported to date.

## Methods

### Ethics statement

This study used deidentified datasets generated during the 4 TB Alliance clinical trials: STAND, Nix-TB, ZeNix and SimpliciTB. Each trial was approved by the applicable local and/ or national Ethics Committee, (S1 Text).

### Statement of informed consent

Written informed consent was obtained from each participant or from their legally authorized representative, parent or guardian, allowing further characterization of the MTB bacteria isolated from the individual.

### MTB isolates and testing scheme

MTB isolates were collected from participants of the 4 trials of pretomanid-containing regimens outlined in Table 1. Sputum collection and primary culture in the BACTEC

**Table 1. Clinical trials and derived analyzed MTB isolates included in this study.**

| Trial Participating countries[1] | Enrollment period | TB drug resistance types[2] | Regimens[3,4] | Participants enrolled | Participants with an isolate with Pa MIC data | | Participants with an isolate with WGS data |
|---|---|---|---|---|---|---|---|
| | | | | | Baseline | Post-baseline[5] | |
| STAND GEO, KEN, MYS, ZAF, TZA, THA, PHL, UGA, UKR | Feb 2015 to Sep 2015 | DS-TB | $6Pa_{200}MZ$ or $4Pa_{200}MZ$ or $4P_{100}MZ$ or 2HRZE/4HR | 271 | 203 (75%) | 22 | 48 (17%) |
| | | RR-TB | $6Pa_{200}MZ$ | 13 | 9 (69%) | | |
| Nix-TB ZAF | Apr 2015 to Nov 2017 | TI/NR MDR-TB, PreXDR-TB, XDR-TB | BPaL | 109 | 57 (52%) | 3 | 56 (51%) |
| ZeNix GEO, MDA, RUS, ZAF | Nov 2017 to Dec 2019 | TI/NR MDR-TB, PreXDR-TB, XDR-TB | $BPaL_{1200x26}$ or $BPaL_{1200x9}$ or $BPaL_{600x26}$ or $BPaL_{600x9}$ | 181 | 143 (79%) | 9 | 143 (79%) |
| SimpliciTB BRA, GEO, MYS, RUS, ZAF, TZA, PHL, UGA | July 2018 to March 2020 | DS-TB | 4BPaMZ or 2HRZE/4HR | 303 | 302 (99.7%) | 9 | 447 (98%) |
| | | DR-TB | 6BPaMZ | 152 | 145 (95%) | | |
| Total | | | | 1029 | 859 (83%) | 43 | 694 (67%) |

[1]BRA = Brazil, GEO = Georgia, KEN = Kenya, MDA = Moldova, MYS = Malaysia, RUS = Russia, ZAF = South Africa, TZA = Tanzania, THA = Thailand, PHL = The Philippines, UGA = Uganda, UKR = Ukraine

[2]Pre-2021 WHO definitions. DR-TB included TB resistant to rifampicin, or isoniazid or both.

[3]B = bedaquiline, E = ethambutol, H = isoniazid, L = linezolid, M = moxifloxacin, Pa = pretomanid, R = rifampicin, Z = pyrazinamide

[4]For regimen details such as dosage and treatment duration, see corresponding trial references.

[5]All patients with an MTB-positive culture at Week 16/17 had their bacteria tested.

Mycobacterial Growth Indicator Tube system™ (MGIT) (Becton & Dickinson) were performed in accordance with study protocols, and select isolates were sub-cultured and shipped to University College London (UCL) Centre for Clinical Microbiology for further characterization for all trials (full details of all procedures conducted at UCL can be found in the Clinical Trial Laboratory Manuals [27]). Baseline isolates grown from any sputum sample collected between Screening and Week 4 of treatment were acceptable, with earlier visits preferred (for 3 participants, pre-screening isolates, obtained from the local TB laboratory were used) (S1 Table). Post-baseline isolates were those grown from sputum collected at or after Month 4 (Week 16 or 17 as per individual study protocols). In some cases, additional isolates from intermediate visits were tested. Post-baseline isolates from participants who had a favorable outcome were also included in the analyses. Generally, the schedules of sputum collection and culture were very similar across the 4 trials (see S1 Text for details).

Prior to additional testing, confirmation of culture purity was obtained using blood agar culture (BAC), acid-fast bacilli microscopy (AFB) and a line probe assay (GenoType Mycobacterium CM or GenoType MTBC [Bruker]). In Nix-TB, ZeNix and SimpliciTB, all MICs were performed using the MGIT method, with the drugs tested determined by the treatment regimen (Table 1). In STAND, MICs for pretomanid and moxifloxacin were performed using the resazurin microtiter plate assay (REMA). All isolates with MIC >cutoff in REMA were retested in the MGIT. Pretomanid MGIT MIC was also performed in a subset of 35 STAND baseline isolates, as part of another study [28]. In addition, MGIT DST was conducted for streptomycin, isoniazid, rifampicin, ethambutol, moxifloxacin and pyrazinamide (plus kanamycin in Nix-TB and ZeNix). WGS performed on baseline and post-baseline isolates allowed paired

comparison to differentiate between relapse and reinfection [29]. The number of participants with baseline and post-baseline isolates included in the study is shown in Table 1, and the results of the DST and MIC tests conducted on the corresponding isolates are listed in S1 Appendix. Across all trials, baseline isolates were not tested for bedaquiline/linezolid for 169, or for pretomanid for 170 participants for various reasons (S1 Table).

### DST and MIC testing

Bedaquiline, linezolid and pretomanid powders were obtained from Janssen, Generon and TB Alliance, respectively. Dimethyl sulfoxide (DMSO) was used to prepare all stock drug solutions and as the diluent for the 2-fold serial dilutions used in the MGIT or REMA MIC determinations (S2 Table). All drug powders were stored in accordance with manufacturers' instructions and prepared stock solutions were frozen at -20˚C in aliquots and stored for up to 6 months, or the expiry date if sooner. Frozen stock solutions were thawed once and discarded afterwards and working solutions were not stored.

MGIT MICs and DST were determined using a MGIT 960 instrument connected to an Epi-Center™ equipped with TBeXIST module (Becton & Dickinson) as described previously [28, 30]. REMA MIC assay was carried out as delineated elsewhere [31], except for inoculum preparation and minor differences detailed in S1 Text. See S1 Fig for the REMA pretomanid MIC distribution from STAND.

Reference strain H37Rv was included in each batch of MIC testing–see S2 Table for acceptable H37Rv ranges. For any tests where the MGIT MIC result was >1 mg/L (≥1mg/L for Nix), or the REMA MIC result was ≥0.5 mg/L, BAC and AFB were done to verify the absence of contamination, and the test was repeated to confirm the result. The WHO recommended critical concentrations were used in DST (S2 Table), and MIC >1 mg/L defined bedaquiline and linezolid resistance, whereas MIC >2 mg/L defined pretomanid resistance [32].

### WGS analyses

Genomic DNA was extracted and sequenced on Illumina sequencing platforms (NextSeq, HiSeq, or MiSeq), at either the Sequencing Laboratory, National Infection Service, Public Health England or the UCL Genomics Unit as per Illumina and local validated protocols (see S1 Text for details). All sequences have been deposited in European Nucleotide Archive (ENA) (S1 Appendix).

WGS data was mapped to the H37Rv reference genome and quality checked as described previously [28] and in the S1 Text. Validated genome sequences underwent the following analyses: (1) identification of variants in genes previously associated with resistance to pretomanid (*ddn*, *fbiA-D*, *fgd1* and *ndh)* or bedaquiline (*mmpR5*, *mmp*, *mmpL5*, *atpB*, *atpE*, *pepQ*, and *Rv1979c*); (2) inspection of the un-translated region (UTR) sequence (plus 100 bp upstream), if any, of those genes to capture mutations in the motif/promoter regions (S3 Table); (3) determination of lineages and key mutations conferring resistance to first and second-line TB drugs, as well as to linezolid, using TB-profiler v.4.1.1 and database tbdb_a9fac19_Feb 16 2022; and (4) construction of 2 maximum likelihood phylogenetic trees (S2 Fig): one for all baselines isolates, the other for baseline/post-baseline samples.

### Calculation of acquired resistance rates

$$x = \frac{a-b}{c-b} \times 100$$

Where,

$a$ = number of participants* with a postbaseline isolate resistant to the drug
$b$ = number of participants* with a baseline isolate resistant to the drug
$c$ = total number of participants*
*Only participants exposed to the drug for $\geq$16 weeks were counted (this excluded early withdrawals, deaths etc.).

## Results

### MTB isolates included in the study

Across the 4 trials analyzed, 859 baseline MTB isolates were tested for susceptibility to at least pretomanid, the drug common to all trials (see Methods for details), with most originating from South Africa (53%), followed by Tanzania (12%) and Georgia (10%) (S1 Appendix; S1 Fig). Additionally, testing for pretomanid was performed on post-baseline isolates from 43 participants who had positive cultures at or after 4 months of therapy (S1 Appendix).

For inclusion in the trials, pre-2021 WHO drug resistance definitions had been used. Further, in Nix-TB and ZeNix, several participants were classified based on their historic, pre-screening DST data (Tables 1–3). Here, we applied the new WHO definitions to reclassify the 694 participants/isolates for which WGS data was available as DS- (47%), RR- (4%), HR- (resistant to isoniazid but not rifampicin) (4%), MDR- (20%), pre-XDR (MDR-TB with additional resistance to a fluoroquinolone) (20%) and XDR-TB (resistant to rifampicin and fluoroquinolones and either bedaquiline or linezolid) (2%), and other resistance (3%).

WGS data was also used to genotype the 694 isolates: lineage 4 (L4) (48%) was the most common genotype, followed by L2 (35%), L3 (9%) and L1 (8%). 1% of samples contained a mixture of lineages and one participant from Georgia harbored *Mycobacterium bovis* (S1 Appendix; S2 Fig).

### Baseline resistance to bedaquiline, linezolid or pretomanid

Overall, 6/859 (0.7%) baseline isolates phenotypically tested were found to be resistant to pretomanid. We observed no marked variation in pretomanid BR rates by trial, TB drug resistance type or enrollment period (Table 2; S1 Appendix). For bedaquiline, however, BR rates

**Table 2. Rates of phenotypically determined baseline resistance to bedaquiline, linezolid or pretomanid resistance across trials.**

| Trial | TB drug resistance types | Baseline resistance rates (%)[1] | | |
|---|---|---|---|---|
| | | **Bedaquiline** | **Linezolid** | **Pretomanid** |
| STAND | DS-TB | *NT* | *NT* | 1/203 (0.5%) |
| | RR-TB | *NT* | *NT* | 0/9 (0%) |
| | Total | *NT* | *NT* | 1/212 (0.5%) |
| Nix-TB | TI/NR MDR-, preXDR- or XDR-TB | 3/58 (5.2%)[2*] | 0/58 (0%) | 0/57 (0%) |
| ZeNix | TI/NR MDR-, preXDR- or XDR-TB | 9/143 (6.3%)* | 2/143 (1.4%) | 3/143 (2.1%) |
| SimpliciTB | DS-TB | 1/302 (0.3%) | *NT* | 1/302 (0.3%) |
| | DR-TB | 0/145 (0%) | *NT* | 1/145 (0.7%) |
| | Total | 1/447 (0.2%)* | *NT* | 2/447 (0.4%) |
| | Overall | 13/648 (2%) | 2/201 (1%) | 6/859 (0.7%) |

*NT* = not tested

*Statistical comparisons of rates in Nix-TB vs. SimpliciTB or in ZeNix vs. SimpliciTB were significant (p-value <0.001).

[1]Details on resistant isolates are provided in Tables 3 and 5.

[2]Bedaquiline BR in Nix-TB was initially reported in a review by Mallick and collaborators [9]. The authors may have missed the fact that only 58 out of the 109 participants had baseline isolates tested, which resulted in a lower rate (2.8%) than the one reported here.

were significantly higher in Nix-TB and ZeNix, as compared to SimpliciTB (5.2–6.3% vs. 0.2%) (Table 2). The key differences between these two groups were the TB drug resistance type and prior TB history. Nix-TB/ZeNix included mostly pre-XDR or XDR patients, who, most often, had previously been extensively treated with other regimens–e.g., in Nix-TB, the mean treatment duration since original TB diagnosis was 23.6 months [24]. Whereas SimpliciTB enrolled only DS- or DR-TB (fluoroquinolone-susceptible) cases who were either newly diagnosed or who had not received anti-TB treatment for 3 years. Prior exposure to bedaquiline for >2 weeks was an exclusion criterion in all 3 trials, but it is possible that previous treatment with clofazimine in the Nix-TB/ZeNix cohorts could have selected for *mmpR5* mutants leading to cross-resistance to bedaquiline. We examined the medical history of all Nix-TB/ZeNix participants and found no correlation between pre-exposure to clofazimine and bedaquiline BR; bedaquiline BR was detected in 3/92 (3.3%, 95%CI 0.7–9.2%) patients with a history of CFZ exposure (>2 weeks) vs. 9/198 (4.5%, 2.1–8.4%) CFZ naïve patients (Table 3; S4 Table).

We identified only 2 cases of linezolid BR, both in ZeNix participants from the Moscow region (ZX103 and ZX121). Their MTB isolates were also resistant to bedaquiline but harbored different resistance mutations and were not epidemiologically linked (Tables 3 and 4; S3 Fig).

Resistance to pyrazinamide or fluoroquinolones (moxifoxacin) were exclusion criteria in STAND, whereas only resistance to fluoroquinolones was an exclusion criterion in SimpliciTB. Pyrazinamide BR in SimpliciTB's DR arm, although common (38% of participants), was not a risk factor for acquisition of resistance to the other study drugs, or for treatment failure/relapse [26]. The only SimpliciTB participant, ST455, with pyrazinamide BR who acquired resistance to another study drug is discussed below.

## Acquired resistance to bedaquiline, linezolid or pretomanid

Consistent with the lower bactericidal activity of Pa-based as compared to BPa-based regimens [33], the trial with the largest proportion of post baseline isolates was STAND (22/284, 7.7%), followed by ZeNix (9/181, 5%), Nix-TB (3/109, 2.7%) and SimpliciTB (9/455, 2%).

Across trials, we observed no case of linezolid AR, and only 3 cases of bedaquiline AR. Among these were participants ZeNix ZX079 and SimpliciTB ST058, who showed poor treatment adherence per clinical records, developed additional resistance to pretomanid, and were ultimately withdrawn. The third participant (NX018) was a confirmed relapse in Nix-TB (Table 4; S2 Fig).

In addition to ZX079 and ST058, 7 other participants developed resistance to pretomanid across trials. Focusing only on ZeNix, bedaquiline BR appeared as a risk for pretomanid AR; 4/9 (44%, 14–79%) participants with bedaquiline BR (including ZX103 and ZX121 with additional linezolid BR) vs. 3/129 (2.3%, 0.5–6.7%) without acquired pretomanid resistance (Tables 3 and 4). Out of these 7 ZeNix participants with pretomanid AR, 5 received linezolid for 9 weeks. None of the Nix-TB participants with bedaquiline BR acquired pretomanid resistance. The only SimpliciTB participant with bedaquiline BR was randomized to the 2HRZE/4HR arm (Table 3).

We also examined the pharmacokinetic data for all participants with any drug AR across trials, looking for signs of unreported non-adherence or poor drug bioavailability. ST455, the only other SimpliciTB participant who acquired pretomanid resistance may have not been fully adherent or, instead, may have been a "fast metabolizer", as evidenced by their low trough concentrations of at least bedaquiline, moxifloxacin and pyrazinamide at various time points (S4 Fig).

## Impact of resistance on treatment outcomes

None of 5 participants who had pretomanid BR and were treated with a pretomanid-containing regimen had an unfavorable microbiological outcome (treatment failure or relapse). In

**Table 3. Participants with baseline resistance to bedaquiline and/or linezolid and/or pretomanid.**

| Trial | Country | Participant | TB history–TB type (duration in days[1]) | Prior treatment with BDQ, CFZ, DLM or LZD (duration in days) | Regimen | Baseline resistance | Acquired resistance | Outcome[2] |
|---|---|---|---|---|---|---|---|---|
| STAND | Ukraine | SD236 | DS (6)[3] | None | 4P$_{100}$MZ | PMD | NA | Favorable; culture converted at Day 84 |
| Nix-TB | S. Africa | NX033 | TI MDR (62)[3] | BDQ (14) | BPaL | BDQ | NA | Favorable; culture converted at Day 16 |
| | | NX083 | MDR (132) XDR (770)[3] | CFZ (722) | BPaL | BDQ | NA | Unfavorable; not TB-related death at Day 76 |
| | | NX099 | MDR (111) XDR (1182)[3] | CFZ (749) | BPaL | BDQ | NA | Favorable; culture converted at Day 57 |
| ZeNix | S. Africa | ZX019 | MDR (16) XDR (10)[3] | CFZ (3), LZD (5) | BPaL$_{1200x9}$ | BDQ | PMD | Unfavorable; confirmed relapse at Day 239 (56 d after last dose) |
| | S. Africa | ZX045 | MDR (283)[3] | None | BPaL$_{1200x9}$ | BDQ, PMD | NA | Favorable; culture converted at Day 29 |
| | S. Africa | ZX056 | XDR (164)[3] | CFZ (133) | BPaL$_{1200x9}$ | BDQ | NA | Favorable; culture converted at Day 14 |
| | Georgia | ZX070 | XDR (73)[3] | None | BPaL$_{1200x26}$ | BDQ | NA | Favorable; culture converted at Day 56 |
| | Georgia | ZX077 | XDR (11)[3] | None | BPaL$_{1200x26}$ | BDQ | NA | Favorable; culture converted at Day 8 |
| | Russia | ZX103 | XDR (105)[3] | None | BPaL$_{1200x26}$ | BDQ, LZD | PMD | Favorable; culture converted at Day 29 |
| | Russia | ZX117 | PreXDR (28)[3] | None | BPaL$_{1200x9}$ | PMD | NA | Favorable; culture converted at Day 8 |
| | Russia | ZX121 | DS (365) MDR (365) XDR (1187)[3] | None | BPaL$_{600x9}$ | BDQ, LZD | PMD | Unfavorable; confirmed relapse at Day 559 (286 d after last dose) |
| | Russia | ZX122 | XDR (29)[3] | None | BPaL$_{600x26}$ | PMD | NA | Favorable; culture converted at Day 15 |
| | Russia | ZX150 | DS (329) XDR (286)[3] | None | BPaL$_{1200x9}$ | BDQ | PMD | Unfavorable; confirmed relapse at Day 188 (last treatment visit) |
| | Russia | ZX152 | DS (2161) MDR (333) XDR (482)[3] | CFZ (5), LZD (5) | BPaL$_{1200x9}$ | BDQ | NA | Unfavorable; withdrawn (AE) during treatment |
| SimpliciTB | Tanzania | ST282 | DS (7)[3] | None | 2HRZE/ 4HR | BDQ | NA | Favorable; culture converted at Day 43 |
| | Tanzania | ST335 | DS (6)[3] | None | 2HRZE/ 4HR | PMD | NA | Unassessable; LTFU, completed treatment and neg status when last seen |
| | Philippines | ST411 | DR (14)[3] | None | 6BPaMZ | PMD | NA | Favorable; culture converted at Day 43 |

*NA* = not applicable (no positive post-baseline isolate)

[1]When the exact date of diagnosis or resolution of the TB episode was unavailable, an approximation to the longest possible duration was used.

[2]Outcomes at the end of the trial: STAND, 18 months after the end of treatment; Nix-TB, 24 months after the end of treatment; ZeNix, 78 weeks (18 months) after the end of treatment; SimpliciTB, 104 weeks (24 months) after the start of treatment.

[3] Ongoing TB episode at the time of enrollment in the respective trial. In both STAND and SimpliciTB, only newly-diagnosed patients or those with no history of TB treatment within 3 years prior to screening were eligible to participate.

Abbreviations: BDQ = bedaquiline, CFZ = clofazimine, DLM = delamanid, LZD = linezolid, LTFU = lost in follow-up, d = days, neg = negative

**Table 4. Participants with no baseline resistance to bedaquiline and/or linezolid and/or pretomanid who acquired resistance to one of these drugs during one of the trials.**

| Trial | Participant | Regimen | Acquired resistance | Outcome[1] |
|---|---|---|---|---|
| Nix TB | NX018 | BPaL | BDQ | Unfavorable; confirmed relapse at Day 267 (83 d after last dose) |
| ZeNix | ZX026 | BPaL$_{600x9}$ | PMD | Unfavorable; withdrawn during treatment; treatment failure |
|  | ZX079 | BPaL$_{600x26}$ | BDQ, PMD | Unfavorable; non-compliant patient who withdrew consent during treatment |
|  | ZX146 | BPaL$_{1200x9}$ | PMD | Unfavorable; confirmed relapse at Day 539 (182 d after last dose) |
| SimpliciTB | ST058 | 6BPaMZ | BDQ, PMD | Unfavorable; withdrawn during treatment; investigator/sponsor decision prompted by patient's poor drug compliance |
|  | ST455 | 6BPaMZ | PMD | Unfavorable; confirmed relapse at Day 222 (42 d after last dose) |

[1]Outcomes and abbreviations are as described in Table 3.

contrast, among the Nix-TB/ZeNix participants, 3/12 (25%, 5–57%) with bedaquiline BR compared to 6/185 (3.2%, 1.2–6.9%) without bedaquiline BR had a negative microbiological outcome (Table 3; S1 Appendix; [25]).

The 2 ZeNix participants with bedaquiline plus linezolid BR (ZX103 and ZX121), despite acquiring resistance to pretomanid, culture converted during the trial at days 29 and 175, respectively. ZX121 later relapsed, whereas ZX103 remained negative until the end of the trial, with a favorable outcome. ZeNix participant (ZX045) with bedaquiline plus pretomanid BR culture converted at Day 29 and remained negative until the end of the trial (Table 3; S3 Fig).

### Genetic characterization of resistant mutants

WGS analyses allowed the identification of several novel mutations in the canonical bedaquiline and pretomanid resistance genes among the resistant isolates (Table 5). Most notably, we observed in 2 isolates (from ZX026 in ZeNix and ST455 in SimpliciTB) large chromosomal deletions including the pretomanid resistant gene *ddn*, along with *loci* previously implicated in cholesterol metabolism and growth *in vivo* (Table 5; S5 Table). All identified mutations conferring resistance to bedaquiline were in *mmpR5*. Also of interest, we were unable to determine the underlying resistance mechanism(s) in 3 bedaquiline-resistant and 4 pretomanid-resistant isolates (Table 5).

A longitudinal analysis of serial sputum cultures from the 10 participants with an AR revealed in some patients a dynamic picture–the apparent turnover of multiple alleles of the

**Table 5. Genetic characterization of isolates resistant to bedaquiline and/or linezolid and/or pretomanid.**

| Trial | Participant | Isolate lineage | Baseline resistance | | Acquired resistance[4] | | Previous report describing mutation, if available |
|---|---|---|---|---|---|---|---|
|  |  |  | Drug (MIC mg/L) | Gene (nucleotide change; amino acid change) | Drug (MIC shift mg/L) | Gene (nucleotide change; amino acid change) |  |
| STAND | SD236[1] | 2.2.1 | PMD (>4) | *ddn* (172C>T; Gln58*) | NA | NA | [34] |
| Nix TB | NX018 | 4.8 | NA | NA | BDQ (0.5→4) | *mmpR5* (139dupG; Asp47fs) | [35] |
|  | NX033 | 4.3.3 | BDQ (4) | *mmpR5* (416T>C; Met139Thr) | NA | NA | [7] |
|  | NX083 | 2.2.2 | BDQ (2) | ND | NA | NA |  |
|  | NX099 | 4.3.3 | BDQ (2) | *mmpR5* (325C>T; Arg109Trp) | NA | NA | [36] |

*(Continued)*

**Table 5.** (*Continued*)

| Trial | Participant | Isolate lineage | Baseline resistance | | Acquired resistance[4] | | Previous report describing mutation, if available |
|---|---|---|---|---|---|---|---|
| | | | Drug (MIC mg/L) | Gene (nucleotide change; amino acid change) | Drug (MIC shift mg/L) | Gene (nucleotide change; amino acid change) | |
| ZeNix | ZX019 | 2.2.2 | BDQ (4) | *mmpR5* (144dupC; Glu49fs) | | | [7] |
| | | | | | PMD (0.25→>16) | *fbiA* (847G>A; Gly283Arg) | Novel mutation |
| | ZX026 | 2.2.2 | *NA* | *NA* | PMD (0.25→>16) | 2.9kb deletion (chromosome position 3982177–4011214)[3] | Novel mutation |
| | ZX045 | Mixed infection, 4.1.1.3 and 2.2.1 | BDQ (4) | *ND* | *NA* | *NA* | |
| | | | PMD (>16) | *ND* | | | |
| | ZX056 | 4.4.1.1 | BDQ (4) | *mmpR5* (201C>G; Ile67Met) | *NA* | *NA* | [7] |
| | ZX070 | 2.2.1 | BDQ (2) | *ND* | *NA* | *NA* | |
| | ZX077 | 2.2.1 | BDQ (2) | *mmpR5* (198dupG; Ile67fs) | *NA* | *NA* | [7] |
| | ZX079 | 2.2.1 | *NA* | *NA* | BDQ (0.25→2) | *mmpR5* (272C>T; Thr91Ile) | [36] |
| | | | | | PMD (0.25→>16) | *fbiC* (557G>A; Gly186Glu) | Novel mutation |
| | ZX103 | 2.2.1 | BDQ (4) | *mmpR5* (136T>G; Cys46Gly) *mmpR5* (64C>T; Gln22*) *mmpR5* (248T>C; Leu83Pro) | | | [7] [36] [12] |
| | | | LZD (>4) | *rrl* (2814G>T; 2270G>T) | | | [37] |
| | | | | | PMD (0.5→>16) | *ddn* (367T>C; Trp123Arg) *fbiC* (989_990dupGC; Ile331fs) *fgd1* (178G>C; Ala60Pro) | Novel mutation Novel mutation Novel mutation |
| | ZX117 | 2.2.1 | PMD (>16) | *ddn* (263G>A; Trp88*) | *NA* | *NA* | [38] |
| | ZX121 | 2.2.1 | BDQ (2) | *mmpR5* (293delA; Asn98fs) | | | Novel mutation |
| | | | LZD (4) | *rplC* (460T>C.; Cys154Arg) | | | [37] |
| | | | | | PMD (0.125→>16) | *ddn* (304A>T; Ile102Phe) *ddn* (352G>A; Glu118Lys) *ddn*(343A>C; Thr115Pro) *fbiC* (1805A>G; Gln602Arg) | Novel mutation [39] Novel mutation Novel mutation |
| | ZX122 | 2.2.1 | PMD (>16) | *ND* | *NA* | *NA* | |
| | ZX146 | 2.2.1 | | | PMD (0.5→>16) | *fgd1* (155delC; Pro52fs) | Novel mutation |
| | ZX150 | 2.2.1 | BDQ (2) | *mmpR5* (198dupG; Ile67fs) | | | [7] |
| | | | | | PMD (0.25→>16) | *fbiA* (35dupG; Ala13fs) *fbiC* (848dupA; Phe284fs) | Novel mutation Novel mutation |
| | ZX152 | 4.8 | BDQ (4) | *mmpR5* (137dupG; Cys46fs) | *NA* | *NA* | [37] |
| SimpliciTB | ST058 | 4.3.4.2.1 | *NA* | *NA* | BDQ (0.5→4) | *mmpR5* (394C>T/Arg132*) | [7] |
| | | | | | PMD (0.125→>16) | *ND* | |
| | ST282 | 4.3.4.2.1 | BDQ (4) | *mmpR5* (65dupT; Arg123fs) | *NA* | *NA* | Novel mutation |
| | ST335[1] | 1.1.2 | PMD (16) | *ddn* (55A>G; Lys19Glu) | *NA* | *NA* | [28] |
| | ST411 | 1.2.1.2.1 | PMD (4)[2] | *ND* | *NA* | *NA* | |
| | ST455 | 4.4.1.1 | *NA* | *NA* | PMD (0.125→>16) | 9.65kb deletion (chromosome position 3923176–4019704)[3] | Novel mutation |

*NA* = not applicable; *ND* = not determined

[1]Isolate described previously [28].

[2]Isolate was tested 3 times and the following MICs were obtained: >1, 4 and 2 mg/L.

[3]For additional information see S5 Table.

[4]For additional information on the relative proportions of each variant listed see S3 Fig.

same or distinct resistant genes–while in others resistance appeared to be linked to the emergence of a single allele of a resistance gene (S3 Fig).

## Discussion

The development of pan-TB regimens, i.e., regimens that could treat both DS- and DR-TB, are based on the idea that combinations of novel anti-TB drugs, with new mechanisms of action and, presumably, little pre-existing resistance, would eliminate the need for DST to determine the appropriate treatment regimen [40]. It is thought that regimens based on bedaquiline, a nitroimidazole (pretomanid or delamanid) and an oxazolidinone (e.g. linezolid or sutezolid) could constitute the backbone of such regimens [3]. However, there is a paucity of information on baseline (pre-existing) resistance to these drugs in most countries, and the impact of these resistances on treatment outcomes. Here we showed that among TB patients enrolled in 4 trials evaluating pretomanid-containing regimens, BR to linezolid, pretomanid and bedaquiline were rare; although, in the case of bedaquiline, a more complex picture emerged.

Only 2 (0.3%) patients were found to be resistant to linezolid (by phenotypic/genotypic DST); both cases also exhibited bedaquiline resistance and were from the Moscow region, although not epidemiologically linked. Dual bedaquiline plus linezolid resistance had been described previously in Russia, but not the specific combinations of resistance mutations reported here [37]. Six (0.7%) out of 859 isolates subjected to MGIT MIC for pretomanid had MIC >2 mg/L, the interim breakpoint recommended by the European Committee for Antimicrobial Susceptibility Testing [32]. These 6 isolates are likely also resistant to delamanid, as cross-resistance between the nitroimidazoles is extensive, although not complete [41]; delamanid was not tested in our study.

Ismail and collaborators reported that 21.1% of patients with previous exposure to bedaquiline or clofazimine vs. 3.6% of those with no previous exposure or unknown previous exposure to these drugs exhibited bedaquiline BR [7]. We found only one case (0.3%) in the SimpliciTB cohort and in the Nix-TB and ZeNix cohorts, rates were 5.2% and 6.3%, respectively. This difference could not be attributed to prior exposure to clofazimine in the Nix-TB/ZeNix participants and agrees with several other studies that identified bedaquiline BR cases among clofazimine and bedaquiline naïve patients [42–44]. It is still unclear what other selective pressures drive bedaquiline/clofazimine resistance emergence linked to *mmpR5* mutations (by far the most common mechanism) in the clinic; exposure to antifungal azoles and/or the antibacterial fusidic acid is a possibility, as overexpression of MmpS5/L5 also confers resistance to these agents [45, 46].

Rates of AR to bedaquiline, linezolid and pretomanid were also low. In fact, none of the 288 Nix-TB/ZeNix participants treated with linezolid acquired resistance to this drug, consistent with the idea that oxazolidinones are the least vulnerable of the classes to resistant mutations [18]. For bedaquiline, we observed only one case of AR in each trial where the drug was included in the regimen, but the cases from ZeNix and SimpliciTB involved participants with adherence or bioavailability issues. The bedaquiline AR rates described here, ranging from 0.4% to 1%, are lower than the ones reported for cohorts of patients treated with regimens containing bedaquiline added to more drugs (median 2.2%, IQR 1.1%– 4.6%) [22]. Lastly, for pretomanid, AR rates were <0.8%, except among ZeNix participants with bedaquiline BR (4/9, 44%). It is possible that in this trial the duration and dose of linezolid may have also played a role, however, the small numbers prevented any meaningful comparison between linezolid dosage arms. In any case, provided they are maintained in programmatic settings–where typically, drug adherence and treatment retention rates are lower, and patient population more

diverse–the overall low AR rates for the 3 drugs we observed are reassuring to efforts to implement BPaL(M) worldwide.

The overall small numbers of BR and AR meant we were unable to perform statistical analyses to determine with confidence risk factors for the emergence of drug resistance and their impact on outcomes. Nevertheless, we made observations that highlight the multifactorial nature of the response to anti-TB treatments and emergence of drug resistance. First, there was no consistent correlation between rates of microbiological failures and rates of AR; in STAND, there were 19 microbiological failures [23], none linked to pretomanid AR; whereas in ZeNix, 5 (2, if one excludes patients with a BR) out of the 7 microbiological failures showed pretomanid AR. Second, BR does not predict treatment outcome. The most striking example of this is the pair of ZeNix participants, ZX103 and ZX121, who exhibited dual bedaquiline/linezolid BR, acquired pretomanid resistance during the trial and yet culture converted. ZX121 later relapsed, whereas ZX103 remained negative until the end of the trial. This difference in response to treatment could be the result, at least in part, of the higher baseline bacterial load of ZX121 (lower MGIT time to positivity) and the slower conversion to negative status (S3 Fig). It is also possible that the combination of resistance mutations acquired by ZX103's post-baseline isolate had a higher fitness cost; or that the difference can be explained exclusively by host factors. PK data did not suggest marked differences in drug metabolism between the 2 participants (S4 Fig).

WGS analyses revealed a diversity of underlying mutations, including novel mutations in canonical resistance genes for bedaquiline and pretomanid/delamanid. In 7/20 (35%) isolates, we were unable to pinpoint the genetic basis of resistance, most likely because there are additional resistant genes yet to be discovered; their identification will probably only come after a few years of programmatic implementation. Our findings together with previous observations that the canonical resistance genes can also exhibit neutral polymorphisms, highlight the challenge of designing low-cost, rapid molecular tests to monitor resistance concurrent with the roll out of the new regimens [47].

Notably, we report chromosomal deletions involving the pretomanid resistance gene *ddn*. Both deletions–one (2.9 kb) in ZeNix and the other (9.65 kb) in SimpliciTB–were observed in isolates from participants who had an unfavorable microbiological outcome. Such large deletions have been previously identified in chromosomal regions including determinants of resistance to streptomycin or pyrazinamide, for example, but remain rare [48]. The deletions described here are also remarkable in that they encompass several genes previously implicated in cholesterol metabolism and growth and persistence *in vivo* (S5 Table). Cholesterol metabolism is now also seen as a target for the development of new anti-TB compounds [4]. Testing the capacity of these mutants to metabolize cholesterol *in vitro* or to grow *in vivo* was beyond the scope of this work.

In summary, our testing of MTB isolates from 859 TB patients (representing a range of disease types) enrolled in trials conducted between February 2015 and March 2020 in 12 countries showed that, overall, rates of pre-existing resistance to linezolid and pretomanid, and to a lesser extent to bedaquiline, were low. Similarly, we noticed overall low rates of emerging resistance to these drugs in the 4 trials, suggesting that BPa-based regimens, BPaL and BPaMZ, effectively protect these drugs. The observations of higher bedaquiline BR among patients with more resistant forms of the disease and higher pretomanid AR among ZeNix participants with bedaquiline BR should trigger further investigation. More work is also needed to understand the genetic basis of these resistances, as well as their impact on treatment outcomes. Along these lines, microbiological data from recently completed and ongoing clinical trials or operational studies of BPa-based regimens may shed light on these points; making this data available

to the TB community should be regarded as a priority. Equally important is to expand the global laboratory capacity to test these drugs.

## Supporting information

**S1 Text. Supplementary methods and references.**
(DOCX)

**S1 Appendix. Participant outcome and microbiological data from the 4 trials.**
(XLSX)

**S1 Table. Disposition of baseline MTB isolates from the 4 trials.**
(DOCX)

**S2 Table. Drug concentrations tested in the MIC and DST analyses.**
(DOCX)

**S3 Table. Genes and upstream regions screened for potential resistance associated variants for pretomanid and bedaquiline.**
(DOCX)

**S4 Table. Participants who were exposed to clofazimine for >2 weeks prior to enrolment and did not show bedaquiline baseline resistance in Nix-TB and ZeNix trials.**
(DOCX)

**S5 Table. Genes completely or partially deleted in pretomanid-resistant isolates from participants ZX026 and/or ST455.**
(DOCX)

**S1 Fig. Pretomanid REMA MIC distribution for all reported STAND baseline isolates (n = 209; blue bars) and the corresponding H37Rv control included with each test run (n = 57; orange bars).** The proposed cutoff for determining resistant isolates based on this data (0.5 mg/L) is indicated by the green line.
(DOCX)

**S2 Fig. Phylogenetic trees.** Phylogenetic tree of **(A)** baseline isolates from all 4 clinical trials for which WGS data available, with lineage shown by the shaded section of the tree. Seven SimpliciTB isolates were excluded from the tree: 6 mixed lineages (ST240, ST336, ST364, ST372, ST400; ZX173) and one *Mycobacterium bovis* (ST200). Phylogenetic tree of **(B)** paired baseline and postbaselines isolates from all participants for which WGS available (n = 34). One participant (NX071) has no corresponding baseline. Numbers on the nodes show SNP differences between isolates. Heat maps depict trial, country of clinical trial site, and drug resistance according to WHO 2021 definitions. Phenotypic resistance to pretomanid, bedaquiline and linezolid by MIC testing is shown as circles (open circle: susceptible; closed circle resistant [based of critical concentrations of 2, 1 and 1mg/L for MGIT MIC respectively, and 0.5mg/L for pretomanid REMA MIC]). Visit abbreviations: S—screening; B—baseline; W–week; M–month; FU–follow up; U–unscheduled; EW–early withdrawal.
(DOCX)

**S3 Fig. Summary data for the 10 participants that acquired resistance to pretomanid and/or bedaquiline.** Tables show the visit schedule, and the corresponding day in the visit schedule relative to the baseline visit as day 0 (S: screening; B: baseline, W: week; M: month, UFU: unscheduled visit in follow up period; EW: early withdrawal). Primary MGIT culture results from the two sputum samples collected at each visit are shown as an overall status from the

visit (MGIT culture +/-), where any positive MGIT results in positive status; and as a mean time to positivity (TTP), represented as days; hours, where two positive MGIT cultures available, or otherwise as the TTP of the single positive MGIT culture (Cx: contaminated/AFB-; Cx +: Contaminated/AFB+/MTB confirmed; ns: no sputum produced). Graphs show MIC data for pretomanid (blue diamond), bedaquiline (green triangle) and linezolid (orange circle) over time, and bars representing % of variants in the resistance genes of interest (see Table 5 and S3 Table) from the WGS data. Dashed lines define the cut-off for resistance. ^ MIC concentration was the highest tested and MIC is greater than the value shown.
(DOCX)

**S4 Fig.** Distributions of plasma concentrations for study drugs in (A) Nix-TB; (B) ZeNix and (C) SimpliciTB for all participants who acquired resistance to pretomanid and/or bedaquiline. Each plot shows the overall distribution of observed concentrations as boxplots, on which are superimposed individual observations for the participants of interest. Pre- and post-dose concentrations were observed at various visits as indicated. Only pre-dose concentrations were observed in Nix-TB (A).
(DOCX)

## Acknowledgments

We are grateful to all trial participants, as well as the staff from the clinical sites and TB laboratories, for their contributions. We thank Jerry Nedelman for sharing the trial participants's PK data and helpful discussions; Matt Betteridge for assistance with the trial databases; and Todd Black and Claudio Köser for critical reading of the manuscript.

## Author Contributions

**Conceptualization:** Juliano Timm, Anna Bateson, Stella Fabiane, Timothy D. McHugh, Eugene Sun.

**Data curation:** Juliano Timm, Anna Bateson, Priya Solanki, Ana Paleckyte, Sylvia A. D. Rofael.

**Formal analysis:** Juliano Timm, Anna Bateson, Adam A. Witney, Sylvia A. D. Rofael, Stella Fabiane.

**Investigation:** Priya Solanki, Ana Paleckyte.

**Methodology:** Juliano Timm, Anna Bateson, Adam A. Witney, Stella Fabiane.

**Project administration:** Juliano Timm.

**Supervision:** Morounfolu Olugbosi, Timothy D. McHugh, Eugene Sun.

**Writing – original draft:** Juliano Timm, Anna Bateson, Timothy D. McHugh.

**Writing – review & editing:** Juliano Timm, Anna Bateson, Adam A. Witney, Sylvia A. D. Rofael, Stella Fabiane, Morounfolu Olugbosi, Timothy D. McHugh, Eugene Sun.

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
