## [Decision Letter · Decision Letter 0]

8 Jun 2023

PGPH-D-23-00590

Baseline and acquired resistance to bedaquiline, linezolid and pretomanid, and impact on treatment outcomes in four tuberculosis clinical trials containing pretomanid

Dear Dr. TIMM,

Thank you for submitting your manuscript to PLOS Global Public Health. After careful consideration, we feel that it has merit but does not fully meet PLOS Global Public Health’s publication criteria as it currently stands. Therefore, we invite you to submit a revised version of the manuscript that addresses the points raised during the review process.

We look forward to receiving your revised manuscript.

Kind regards,

Raquel Muñiz-Salazar, Ph.D.

Academic Editor

Journal Requirements:

Additional Editor Comments (if provided):

Both reviewers have suggested improvements to the manuscript. Therefore major revisions are required.

Reviewers' comments:

Reviewer's Responses to Questions

**Comments to the Author**

1. Does this manuscript meet PLOS Global Public Health’s publication criteria? Is the manuscript technically sound, and do the data support the conclusions? The manuscript must describe methodologically and ethically rigorous research with conclusions that are appropriately drawn based on the data presented.

Reviewer #1: Yes

Reviewer #2: Yes

2. Has the statistical analysis been performed appropriately and rigorously?

Reviewer #1: No

Reviewer #2: N/A

3. Have the authors made all data underlying the findings in their manuscript fully available (please refer to the Data Availability Statement at the start of the manuscript PDF file)?

Reviewer #1: Yes

Reviewer #2: Yes

4. Is the manuscript presented in an intelligible fashion and written in standard English?

Reviewer #1: Yes

Reviewer #2: Yes

5. Review Comments to the Author

Reviewer #1: Overview:

In this manuscript, Timm et al. present data from four pretomanid-containing clinical trials – STAND, Nix-TB, ZeNix and Simplici TB to investigate the rates of baseline resistance (BR) and acquired resistance (AR) to bedaquiline (B), pretomanid (Pa) and linezolid (L) (BPaL) drugs. They also present data on the genetic basis, risk factors and impact on treatment outcomes. While linezolid and pretomanid BR rates were rare, they observed a relatively higher rate of BR in bedaquiline in patients with highly resistant TB or patients with longer prior treatment histories. Additionally, they report that BR was a risk factor for bacteriological failure or relapse in Nix-TB/ ZeNix participants. Overall, they conclude that the low rate of BR to linezolid and pretomanid are in support of the worldwide implementation of BPaL-based regimens.

Comments:

1. An introduction about the global burden of TB should be provided in the beginning of the introduction. Adding a few sentences on what has been the shortcomings of the regimen prior to BPaL (INH, rifampicin) should be provided to then pave way for the need to include BPaL – based regimens.

2. Line 44 – which specific populations/ epidemiological regions has WHO enlisted in the new regimens should be mentioned for readers not familiar with the details.

3. Line 45 states “in clinical trials” but only cites a single reference at the end of the sentence.

4. Line 47 – please mention which combination therapies it was approved as part of.

5. Lines 50 – 53 – please provide adequate background information on published literature on Mtb drug resistance before introducing the biochemistry details.

6. Line 53: No reference.

7. Line 55: what type of Mtb cells? No reference.

8. Line 56: What is the rational of mentioning clofazimine?

9. Overall, the introduction needs to be improved in making the connecting dots in between change of paragraphs. In its current format, each paragraph is a stand-alone entity and fails to provide a connected picture of the study.

10. Line 77 – 81 is too long and confusing to read.

11. Line 83: The authors should provide a reason and references to substantiate this outcome of measuring only bedaquiline resistance and in small number of countries as compared with the other two drugs?

12. Table 1: Is it possible to provide the timelines for post-baseline measurements in these cohorts?

13. Is there a reference that can be added at Line 121?

14. Can the authors state how comparable were the time points of sputum collection between the 4 cohorts?

15. Lines 127 – 129: Please provide references for the protocols used.

16. How do the authors factor in the variation (if any) from the different MIC assays used in the cohorts? How do they factor that user variations when retesting the samples? What controls were used?

17. Why were only 694 isolates genotyped out of a total of 859 baseline Mtb isolates? Was there any particular cohort that had the majority of baseline WGS data missing? Does this present a confounding factor in this type of comparative analysis?

18. Line 83: Did the authors look deeper into other factors to see why this variation in susceptibility to pretomanid existed in different locations?

19. Line 240: How long after treatment completion did the participant relapse?

20. Table 4 lists the participants who acquired resistance with no basline resistance to bedaquiline and/or linezolid and/ or pretomanid. However, 50% of the participants listed were withdrawn from the treatment due to different reasons. It is not clear what the take home message here really is?

21. Line 245: It might be a little premature to link bedaquiline resistance as a risk for pretomanid AR unless further evidence exists to link the two mechanistically.

22. Line 258: True but what about longer term relapse?

23. Could the authors mention the fitness levels of the Mtb isolates from these cohorts and if that had a role to play in the mutation rates observed?

24. Lines 274 – 277: Were these observations made in different cohorts or same cohorts? Does the epidemiology or treatment regimens play a role?

25. How translational are the findings when applied to a larger cohort given the small numbers of resisters and no statistical significance observed?

Reviewer #2: PGPH-D-23-00590

Research Article: Baseline and acquired resistance to bedaquiline, linezolid and pretomanid, and impact on treatment outcomes in four tuberculosis clinical trials containing pretomanid

Summary

This paper is describing pre-existing and acquired resistance on combined from 4 clinical trials with novel drugs. The material is presented well, and the findings are interesting, original and important. The paper is comprehensive presenting data not only in the context of the cohort studied but also down to the individual that developed resistance.

Major comments

An important message from the paper’s main findings should be made more clearly. Resistance rates were rare, even in patients highly resistant to other agents, there was great variation in the underlying mutations, mot all mutations found during in vitro testing were clinically relevant, and in quite a few cases the reason for resistance could not be found at all (T5). This shows how difficult, if not impossible, it would be to have rational methods for resistance testing rolled out at low cost together with novel drugs and regimens. It would not be possible to rationally design such tests without the experience of a few years of usage in the field. This should be added to the discussion prominently.

Minor comments

Abstract

Add percent to every number.

L37 The conclusion about the protection of drugs is valid, however, it must also be considered that there is more than just a drug combination that avoids resistance at play. Clinical trials for drug or regimen registration will ensure that underdosing or poor compliance are avoided. Also, the study population is selected and looked after with much larger resources than available upon implementation in the field (this goes for a few other places where this claim is made).

Introduction

L45 and other places: In the context of these novel drugs and regimens I would recommend not to use definitions that might in a few years be obsolete. DS, DR, RR, MDR, pre-XDR etc are concepts based on currently available regimens that will hopefully soon be replaced with better ones. I would recommend, where possible, to name the agents that this refers to, and remain conscious about how this will read to a person in 10- or 20-years’ time. At some point it will no longer be intuitive what is meant by DS-TB and DR-TB.

L69 Consider replacing production with synthesis.

L78 Introduce delamanid above under nitroimidazoles.

L81 I disagree that it’s only the limited capacity. There is also our limited knowledge what to look for, which is the very reason the research was done that is reported here.

L97 instead of 7 years say from xxxx to xxxx

Methods

T1 and L139 – I was a bit confused about if the is 170 (as per table totals 1029 minus 859) or 169 as per text. This won’t make a major difference but please check that this is all correct and clear.

L125 – any participant that still has an isolate at M4 is at high risk of having a poor outcome. Perhaps rewrite what is meant with “regardless of …”

Results

Throughout, also on the tables, present % not only numbers.

T2 L204. Perhaps find a more diplomatic way to explain what those authors did, which was also passed by reviewers and editors. There must be a good reason to do it the way they did, even if these authors disagree.

L206/207: Some of this text can be shortened.

L213 – how was this history examined? This is not quite in line with the claim that the data was anonymized for analysis. What information was available and how was it accessed?

T3: It is interesting that there was not a single patient without previous TB, i.e., that was de-novo infected with a resistant strain. So no transmission of resistant TB was observed in this cohort. This is a bit in contrast with some claims that transmission of resistant TB is a driver of resistance in the field. Can this somehow be explained with the population selection, or a different theory?

Discussion

This is generally a bit lengthy and often repeating what has already been nicely presented. It should be shortened and focused on what the findings mean rather than repeating them.

6. PLOS authors have the option to publish the peer review history of their article (what does this mean?). If published, this will include your full peer review and any attached files.

**Do you want your identity to be public for this peer review?** For information about this choice, including consent withdrawal, please see our Privacy Policy.

Reviewer #1: **Yes: **Riti Sharan

Reviewer #2: **Yes: **Andreas Diacon

---

## [Decision Letter · Decision Letter 1]

18 Sep 2023

Baseline and acquired resistance to bedaquiline, linezolid and pretomanid, and impact on treatment outcomes in four tuberculosis clinical trials containing pretomanid

PGPH-D-23-00590R1

Dear Dr. TIMM,

We are pleased to inform you that your manuscript 'Baseline and acquired resistance to bedaquiline, linezolid and pretomanid, and impact on treatment outcomes in four tuberculosis clinical trials containing pretomanid' has been provisionally accepted for publication in PLOS Global Public Health.

Best regards,

Raquel Muñiz-Salazar, Ph.D.

Academic Editor

Taking the reviewers' comments, I recommend ACCEPT IT.

Reviewer Comments (if any, and for reference):

Reviewer's Responses to Questions

**Comments to the Author**

1. If the authors have adequately addressed your comments raised in a previous round of review and you feel that this manuscript is now acceptable for publication, you may indicate that here to bypass the “Comments to the Author” section, enter your conflict of interest statement in the “Confidential to Editor” section, and submit your "Accept" recommendation.

Reviewer #2: All comments have been addressed

2. Does this manuscript meet PLOS Global Public Health’s publication criteria? Is the manuscript technically sound, and do the data support the conclusions? The manuscript must describe methodologically and ethically rigorous research with conclusions that are appropriately drawn based on the data presented.

Reviewer #2: Yes

3. Has the statistical analysis been performed appropriately and rigorously?

Reviewer #2: Yes

4. Have the authors made all data underlying the findings in their manuscript fully available (please refer to the Data Availability Statement at the start of the manuscript PDF file)?

Reviewer #2: Yes

5. Is the manuscript presented in an intelligible fashion and written in standard English?

Reviewer #2: Yes

6. Review Comments to the Author

Reviewer #2: I would like to thank the authors for their diligence in answering the queries and adjusting the manuscript.

7. PLOS authors have the option to publish the peer review history of their article (what does this mean?). If published, this will include your full peer review and any attached files.

**Do you want your identity to be public for this peer review?** For information about this choice, including consent withdrawal, please see our Privacy Policy.

Reviewer #2: **Yes: **Andreas Diacon
